# Study of Archaeal Diversity in the Arctic Meltwater Lake Region

**DOI:** 10.3390/biology12071023

**Published:** 2023-07-20

**Authors:** Yiling Qin, Nengfei Wang, Li Zheng, Qinxin Li, Long Wang, Xiaoyu Xu, Xiaofei Yin

**Affiliations:** 1First Institute of Oceanography, Ministry of Natural Resources, Qingdao 266061, China; qinyl2021@126.com (Y.Q.); zhengli@fio.org.cn (L.Z.); yxf@fio.org.cn (X.Y.); 2School of Chemistry and Chemical Engineering, Linyi University, Linyi 276000, China; xiaoyuxu_xxy@163.com; 3College of Chemistry and Chemical Engineering, Qingdao University, Qingdao 266071, China; liqinxina@163.com; 4Department of Bioengineering, College of Marine Sciences and Biological Engineering, Qingdao University of Science & Technology, Qingdao 266042, China; wanglong1505108440@163.com

**Keywords:** Arctic Ny-Ålesund, meltwater lakes, high-throughput sequencing, archaea diversity, soil physicochemical properties, WGCNA

## Abstract

**Simple Summary:**

The Arctic is experiencing a significant temperature increase under the global warming trend. As a result, the Arctic permafrost is thawing, glacial meltwater is gathering, and the depressions are gradually forming lakes, affecting the composition and material cycles of the terrestrial ecosystems. Two meltwater lakes with different landscapes in the Ny-Ålesund region of the Arctic were taken as study objects. The structure of the archaeal community and the influence of soil physiochemical factors on the archaeal community were investigated. The differences in the archaeal community structure between the intertidal and subtidal zones of the two lakes were compared, and the reasons for the differences were analyzed and discussed. A redundancy analysis identified NH_4_^+^, SiO_3_^2−^, MC, NO_3_^−^, and NO_2_^−^ as key soil physicochemical factors that have significantly influenced the structure of the archaeal community. The hub archaea in the archaeal community were identified by weighted gene co-expression network analysis (WGCNA). The use of WGCNA in this study provides new ideas for future research on the microbial community structure. In the context of global warming, this study contributes to research on archaeal communities in Arctic meltwater lakes in response to climate change.

**Abstract:**

Two typical lakes formed from meltwater in the Ny-Ålesund area were taken as the study subjects in 2018. To investigate the archaeal community compositions of the two lakes, 16S rRNA genes from soil samples from the intertidal and subtidal zones of the two lakes were sequenced with high throughput. At the phylum level, the intertidal zone was dominated by Crenarchaeota and the subtidal zone was dominated by Halobacter; at the genus level, the intertidal zone was dominated by *Nitrososphaeraceae_unclassified* and Candidatus_*Nitrocosmicus*, while the subtidal zone was dominated by *Methanoregula*. The soil physicochemical factors pH, moisture content (MC), total organic carbon (TOC), total organic nitrogen (TON), nitrite nitrogen (NO_2_^−^-N), and nitrate nitrogen (NO_3_^−^-N) were significantly different in the intertidal and subtidal zones of the lake. By redundancy analysis, the results indicated that NH_4_^+^-N, SiO_3_^2−^-Si, MC, NO_3_^−^-N, and NO_2_^−^-N have had highly significant effects on the archaeal diversity and distribution. A weighted gene co-expression network analysis (WGCNA) was used to search for hub archaea associated with physicochemical factors. The results suggested that these physicochemical factors play important roles in the diversity and structure of the archaeal community at different sites by altering the abundance of certain hub archaea. In addition, *Woesearchaeales* was found to be the hub archaea genus at every site.

## 1. Introduction

Under the global warming trend, the Arctic region is one of the most obvious regions affected by the temperature rise in the last hundred years [1]. The surface temperature in the Arctic is changing more than twice as fast as the global warming average, which means that there is an “Arctic amplification effect” [2]. As Arctic temperatures continue to rise, depressions created by thawing permafrost and then the melting of glaciers gradually form lakes, which leads to changes in terrestrial ecosystems and material cycles [3,4]. Lakes are an important part of the Arctic environment, and climate change has had a significant impact on Arctic lakes, with repeated freezing and thawing directly affecting their abundance and distribution [5]. The Svalbard lake survey showed that meltwater lakes are one of the major lake types, accounting for 30.4% of the lake area [6].

Changes in the abundance and distribution of lakes in the Arctic cause changes in the soil water content, nutrients, microbial diversity, and ecosystem function [5]. In Arctic soils, the microbial diversity is rich and dominates as decomposers in the terrestrial environment of the polar regions [7]. Changes in some key elements of the soil can cause changes in the original biogeochemical cycle, which can significantly alter the structure of biological communities as well as the ecosystem function [8]. Archaea are important microorganisms that play important roles in the material cycling processes of geochemistry, such as carbon cycling and nitrogen cycling processes [9]. The community structure and distribution of archaea are influenced by various physicochemical factors, such as the temperature [10], pH [11], and nitrogen [12], phosphorus [13], oxygen [14], soil moisture, and organic matter contents [15].

Microbial studies of Arctic lakes have mainly focused on bacteria with few studies conducted on archaea. The research on archaea in the Arctic Svalbard region has focused on marine sediments [16], glaciers [17], tundra [18], and Arctic fauna [19]. In the few studies of archaea in Svalbard lakes, some researchers have investigated the vertical structure of lake archaeal communities [20], but there have been no relevant studies on the structural differences between intertidal and subtidal archaeal communities. It is clear that the archaea in the lakes in the Ny-Ålesund area have been poorly and incompletely studied. Not only the structure and differences in the archaeal community composition in the lake sediments but also the correlation between the archaeal communities and relevant environmental factors have not been investigated. In addition, there are complex interrelationships among microorganisms, such as competition, symbiosis, and parasitism [21], and the specific forms are competitive substrates [22], direct electron transfer between species [23], and complementary metabolites [24].

In this study, intertidal and subtidal soil samples were collected from two lakes with different characteristics in the Ny-Ålesund area to reveal the structure and differences in soil archaeal communities and to explore the role and influence of environmental physicochemical factors on the structure and diversity of the archaeal communities. The archaeal diversity composition of the soil samples was analyzed by high-throughput 16S rRNA sequencing. At the same time, the physicochemical properties of the soil samples were determined by conventional instrumental analysis to further study their effects on the structure of soil archaeal communities and to identify the most important influencing factors. WGCNA was then used to find the hub that influences the archaeal community’s composition.

## 2. Materials and Methods

### 2.1. Study Site Description and Sample Collection

The study site is the Ny-Ålesund area, which is located in the Svalbard archipelago of the Arctic. Svalbard is located at the intersection of the northern North Atlantic Ocean and the Arctic Ocean basin, and therefore, the climate there differs significantly from the rest of the Arctic [25,26]. Ny-Ålesund is located on the western coast of Spitsbergen, which is the largest island in Svalbard. Due to its proximity to, and influence from, the warm Western Spitsbergen Current, the climate of Ny-Ålesund is generally warmer and wetter than that of other land masses at the same latitude [27]. Samples were collected in July 2018 from 12 sites in the Ny-Ålesund region of the Arctic (Table 1) with three parallel samples set at each site, giving a total of 36 soil individual samples. Figure 1 shows the locations of the two lakes on the map. The intertidal zone refers to the area that used to be above water, but which became submerged as meltwater increased due to warmer temperatures. The subtidal zone refers to the area that is submerged throughout the year. Approximately 50 g of soil was dug up with a sterile spoon into sterile bags at a depth of approximately 5 cm. At the end of sampling, the samples were stored at −20 °C immediately and promptly transferred to an ultra-low temperature refrigerator (−80 °C).

### 2.2. Physicochemical Properties of Soils

Nine physicochemical factors of the soil samples were measured, among which were the pH, moisture content (MC), total organic carbon (TOC), total organic nitrogen (TON), and the concentrations of five soluble nutrients: nitrite nitrogen (NO_2_^−^-N), nitrate nitrogen (NO_3_^−^-N), ammonium nitrogen (NH_4_^+^-N), silicate silicon (SiO_3_^2−^-Si), and phosphate phosphorus (PO_4_^3−^-P). The pH of the soil was measured with a pH meter (PHS-3C, Shanghai REX Instrument Factory, Shanghai, China) by taking 4 g of the soil sample and adding 10 mL of distilled water. The MC was measured by weighing 10 g of the soil samples, drying at 105 °C to a constant weight, and calculating the weight loss. Soil samples were freeze-dried immediately after removal from the −80 °C refrigerator for the determination of other physicochemical factors. The freeze-dried samples were decalcified with hydrochloric acid, rinsed with deionized water, and dried before being analyzed for TOC and TON using an element analyzer (EA30000, Euro Vector SpA, Milan, Italy) [28]. The contents of the five soluble nutrients were determined using freeze-dried samples treated with deionized water at a ratio of 1:10 (g∙mL^−1^) and analyzed by an automated nutrient analyzer (QuAAtro, SEAL, Germany) with a relative standard deviation of less than 0.5%. One-way ANOVA and Duncan’s test were performed on the obtained physicochemical factors’ data using Origin 2021.

### 2.3. DNA Extraction, PCR Amplification and Sequencing

Total DNA extraction from the soil was performed according to the Mo Bio Power SOIL DNA isolation kit manufacturer’s instructions, and the DNA obtained was tested for DNA purity and integrity in agarose gel. DNA at concentrations of 10–30 ng/μL^−1^ was used for PCR amplification with Arch519F (5′-CAGCCGCCGCGGTAA-3′) and Arch915R (5′-GTGCTCCCCCCCGCCAATTCCT-3′) [29] as primers. The V4-V5 region of the 16S rRNA gene was amplified according to the method described by He et al. [15], and PCR products of 400–450 bp were recovered with a gel extraction kit and sent to a commercial sequencing company for sequencing.

### 2.4. High-Throughput Sequencing and Statistical Analysis

The 16S rRNA gene V4-V5 region was sequenced with high throughput on the Illumina Miseq platform [30], and low-quality sequences were removed by quality control and cascading in order to obtain clean data. Original read segments were submitted to the NCBI Sequence Read Archive database (accession number: PRJNA956435). We performed quality control on the raw data, which included data splicing, filtering, and chimera removal processing. The DADA2 method in the QIIME 2.0 system was used for noise reduction to perform dereplication or the equivalent to form clustering at 100% similarity [31]. The ASVs (Amplicon Sequence Variants) were each de-duplicated sequence (corresponding to OTU representative sequences) obtained after noise reduction. Species annotations were made for each ASV by the classify-sklearn algorithm in QIIME 2.0 using Silva 138.1 [32]. Statistical analysis of the diversity indices of ASVs was performed using the QIIME 2.0 system. The QIIME2 system was used to calculate the observed_otus, shannon, simpson, chao1, goods_coverage, dominance, and pielou_e indices. A significantly different species analysis between groups was performed using LEFSE in the QIIME2 system. A redundancy analysis (RDA) was conducted using R software (version 4.0.5) to explore the correlation between soil physicochemical properties and the distribution of the archaeal community. The RDA analysis started with a multiple regression of the matrix of the species composition and the matrix of the environmental variables, and then the fitted value matrix was obtained. PCA was performed on the fitted value matrix to obtain the canonical eigenvector matrix, and then the image was plotted. The weighted gene co-expression network analysis (WGCNA) software package was used to differentiate the modules [21]. Then, the heatmap of modules and physicochemical factors and the network map of the core archaea were drawn. The results of the WGCNA were combined with the results of the RDA to analyze the influence of soil physicochemical factors on the structure of the archaeal community.

## 3. Results

### 3.1. Physicochemical Properties of Soil Samples

Nine physicochemical properties, including the pH, MC, TOC, TON, NO_2_^−^-N, NO_3_^−^-N, NH_4_^+^-N, SiO_3_^2−^-Si, and PO_4_^3−^-P, were measured in 36 soil samples from 12 sites. As can be seen from Table 2, except for NH_4_^+^-N, SiO_3_^2−^-Si, and PO_4_^3−^-P, other physicochemical factors were significantly different in the intertidal and subtidal zone of the lake. The contents of NO_2_^−^-N and NO_3_^−^-N in NHS were significantly higher than that of NHX, the content of NH_4_^+^-N was slightly higher, while PO_4_^3−^-P was the opposite. The pH of XHS was significantly higher than that of XHX, while the TOC was reversed. The contents of NO_2_^−^-N and NO_3_^−^-N in XHX2 and XHS3 were significantly higher than those in other places. Among them, XHX2 had the highest content of SiO_3_^2−^-Si, and XHX3 had the highest contents of NH_4_^+^-N and PO_4_^3−^-P.

### 3.2. Diversity and Structure Analysis of Archaeal Community

A total of 36 samples from two lakes in Ny-Ålesund, Arctic, were sequenced with high throughput, and 1,960,571 archaeal sequences were obtained, with an average of 54,460 archaeal sequences per sample. Clustering by the DADA2 method yielded 1906 ASVs for subsequent analysis. After the alpha diversity analysis, the values of Good’s coverage in all sites were greater than 0.9999, indicating that the coverage of sequencing was high enough, and the vast majority of sequence regions were obtained by sequencing.

As can be seen from Figure 2a, except for the unclassified ones, all samples of ASVs were clustered into seven phyla. Among them, the highest relative abundance was found for Halobacterota, followed by Crenarchaeota, and these two phyla accounted for a much higher proportion of all sites than other phyla. However, in general, the distributions of the two phyla in intertidal and subtidal soils differed significantly, with Crenarchaeota predominating in the intertidal zones and Halobacterota predominating in the subtidal zones. Additionally, the relative abundance of Euryarchaeota in the intertidal and subtidal samples also differed significantly, as it was more abundant in subtidal zones than in intertidal zones.

In Figure 2b, it can be seen that the relative abundance of each genus at each sampling site differed significantly. In general, the distributions of archaea genera in two lakes were significantly different, and the dominant genera in the lake with birds were less significant than in the small lake. For example, *Nitrososphaeraceae_unclassified*, *Methanoregula*, and Candidatus_*Nitrocosmicus* had high relative abundances in the small lake. There were also differences between the intertidal and subtidal zones with high relative abundances of *Nitrososphaeraceae_unclassified* and Candidatus_*Nitrocosmicus* in the intertidal zones and *Methanoregula* and *Methanosaeta* in the subtidal zones.

### 3.3. Correlation between Soil Physicochemical Factors and Archaeal Community Structure

RDA was used to explore the relationships among nine soil physicochemical factors and archaeal communities. As shown in Figure 3, the first two axes explained 40.42% of the total variation in the structure of the archaeal community. The results showed that NH_4_^+^, SiO_3_^2−^, MC, NO_3_^−^, and NO_2_^−^ had highly significant effects on the archaeal diversity and distribution. Among all sampling sites, NH_4_^+^, SiO_3_^2−^, and MC had the greatest effects on XHX2, while NO_3_^−^ and NO_2_^−^ had the greatest effects onXHS3. Then, the Monte Carlo permutation test was applied, and the results were as follows (Table 3): NH_4_^+^ (*r^2^* = 0.4399, *Pr* = 0.001), SiO_3_^2−^ (*r^2^* = 0.4911, *Pr* = 0.001), MC (*r^2^* = 0.4264, *Pr* = 0.001), NO_3_^−^ (*r^2^* = 0.4254, *Pr* = 0.001), and NO_2_^−^ (*r^2^* = 0.5296, *Pr* = 0.001). The results further verified the significance of the effects of these five soil physicochemical factors on the archaeal community. In addition, TOC and pH had significant effects on the diversity and distribution of archaeal communities.

We not only studied the physicochemical factors but also applied LEFSE to focus on archaea with significant large differences in the relative abundance at each site. The results are shown in Figure 4a. There were significant differences in archaea at nine of the twelve studied sites. For genera with significant differences, NHX2 had the most differential archaea, followed by NHX3, and then XHS3 and NHX3. The LEFSE results for the two lakes are shown in Figure 4b with five distinctly different archaeal genera in NH (lake with birds) and XH (small lake) identified.

### 3.4. The Weighted Gene Co-Expression Network Analysis

The weighted gene co-expression network analysis (WGCNA) method was used to analyze the relationship between soil physicochemical factors and all ASVs annotated clearly as archaea, and the results are presented as a heatmap of the modules and physicochemical factors (Figure 5). The turquoise module showed a highly significant positive correlation with PO_4_^3−^ (*r* = 0.85, *p* < 0.01), NH_4_^+^ (*r* = 0.68, *p* < 0.01), and TOC (*r* = 0.57, *p* < 0.01). The gray module also showed a significant positive correlation with NH_4_^+^ (*r* = 0.61, *p* < 0.01) and significant negative correlations with NO_3_^−^ (*r* = −0.48, *p* < 0.01), NO_2_^−^ (*r* = −0.46, *p* < 0.01), and TON (*r* = −0.51, *p* < 0.01). The results of the turquoise module were imported into Cystoscope and plotted as a network diagram (Figure 6). As can be seen from the figure, there were 19 hub archaea groups, including three genera, *Woesearchaeales*, Candidatus_*Nitrososphaera*, and *Methanobacterium*, which were mainly *Woesearchaeales*. The network diagram of the gray module can be seen in Figure 7. The results show that the hub archaea differed among sites, but the hub genera all included *Woesearchaeales*. The hub genera in NHX2 were *Woesearchaeales* and *Methanosaeta*, the hub genera in XHS were *Woesearchaeales* and Candidatus_*Nitrososphaera*, and the hub genera in XHX3.1 and XHX3.2 were *Woesearchaeales* and *Methanoregula*, while *Woesearchaeales*, Candidatus_*Nitrososphaera* and *Methanobacterium* were identified as the hub genera in XHX3.3.

## 4. Discussion

This study revealed the composition and diversity of archaeal communities in a typical Arctic lake area and analyzed the differences between intertidal and subtidal areas. The results indicate that soil physicochemical factors influence the archaeal community composition. In this study, 1906 archaeal ASVs were identified with Good’s coverage values exceeding 99.99% at each sampling site. Although not as diverse as the bacterial communities in similar meltwater areas [33], richly diverse archaeal communities were found when compared to previous archaeal studies in the Svalbard region [34]. The 16S rRNA high-throughput sequencing data showed that the phyla Halobacterota and Crenarchaeota were absolutely dominant, and there were significant differences in the intertidal and subtidal zones. Halobacterota was reported as the major archaeal phylum in extreme environments, such as mud volcanoes [35], Antarctic salt cones [36], and deserts [37], as it can mostly tolerate saline environments but is less tolerant to oxygen and desiccation [38]. Oxygen levels and aridity are higher in the intertidal zone than in the subtidal zone, so the abundance of Halobacterota archaea is higher in the subtidal zone than in the intertidal zone. It has been shown that Halobacterota has sulfate allosteric reduction genes involved in the sulfur cycle, and some genes participate in the carbon cycle, such as those of the order Methanomicrobiales [39,40]. Crenarchaeota has been reported as the major archaeal phylum in both Antarctic and Arctic archaeal studies [41,42], and it was also the major archaeal phylum in previous Svalbard archaeal studies [34]. This is a diverse and widespread phylum that is mainly characterized by acidophilic, thermophilic, and anaerobic properties [43]. It also contains ammonia-oxidizing archaea, which are important players in the natural nitrogen cycle [44]. In the genus classification, the dominant genera were *Nitrososphaeraceae_unclassified*, *Methanoregula*, and Candidatus_*Nitrocosmicus*. The archaeal community structures in the intertidal and subtidal zones were significantly different, and those in the two lakes were also significantly different.

In the Arctic, soil physicochemical factors influence the diversity of archaeal communities and show significant correlations with the archaeal community composition. In this study, there were differences in the physicochemical properties of the soils that had different effects on the archaeal communities at different sampling sites. According to the results of the RDA, NH_4_^+^, SiO_3_^2−^, MC, NO_3_^−^, and NO_2_^−^ were highly correlated with the archaeal community composition of all samples. In general, these five physicochemical factors had greater influences on the small lake (XH) than on the lake with birds (NH). However, the different physicochemical factors affected the intertidal and subtidal zones to different extents. MC was significantly higher in the subtidal than in the intertidal zone, while the opposite was true for NO_3_^−^ and NO_2_^−^. pH and TOC differed significantly only between XHS and XHX, while NH_4_^+^, SiO_3_^2−^, and PO_4_^3−^ were high at some sites but not at most. These differences may result in differences in the composition of archaeal communities, while at the same time, the archaeal communities may also counteract the physicochemical properties of the soil [45]. In addition, global warming has affected the abundance and distribution of Arctic meltwater lakes which, in turn, has led to changes in the soil water content, nutrients, and microbial diversity [5]. In this study, the water content had a significant effect on the archaeal community. Therefore, the archaeal communities of the lakes might be affected by changes in the meltwater volume due to climate change. There was a study that showed that Arctic peat archaeal communities were dominated by nonmethanogenic archaea at lower temperatures, and the diversity and relative abundance of methanogenic archaea increased at higher temperatures [10]. Research has shown that glaciers have unique archaeal community physicochemical factors [46], so the glacial meltwater input may affect the local archaeal communities of lakes. These changes may show the response of the archaeal community to environmental changes.

The dominant genera in both lakes were mainly *Nitrososphaeraceae* (unclassified) and Candidatus_*Nitrocosmicus* in the intertidal zones and *Methanoregula* and *Methanosaeta* in the subtidal zones. The difference in dominant genera between the two regions may be related to the significantly higher NO_3_^−^ and NO_2_^−^ concentrations in the intertidal zones than in the subtidal zones. *Nitrososphaeraceae* belongs to the phylum Crenarchaeota and produces energy by oxidizing ammonia to nitrite under aerobic conditions [47]. *Nitrososphaeraceae* species are involved in the soil nitrogen cycle and play an important role [48], which explains the higher abundance of *Nitrososphaeraceae* in the intertidal than in the subtidal zones. Candidatus_*Nitrocosmicus* also belongs to the phylum Crenarchaeota and is a genus in the family Nitrososphaeraceae. It has a strong aerobic ammonia oxidation biofunction and can oxidize high concentrations of ammonia [49,50]. As observed in both Figure 2b and Figure 4a, the relative abundance of the genus Candidatus_*Nitrocosmicus* at XHS3 was significantly higher than that at the other sites, corresponding to the apparently high NO_3_^−^ and NO_2_^−^ concentrations at this site. *Methanoregula* and *Methanosaeta* are both members of the phylum Halobacterota, but in different classes. *Methanoregula* belongs to the class Methanomicrobia and is a strictly anaerobic, thermophilic organism that produces methane from H_2_, CO_2_ or formate [51,52]. Global distribution studies of sequences from this genus indicate that this lineage survives in a variety of environments [53], and some species tend to grow in nitrogen-rich environments [54]. This partly explains the significantly higher relative abundance of *Methanoregula* at XHX2 and XHX3, which have significantly higher NH_4_^+^ contents, than at other sites. *Methanosaeta* belongs to the class Methanosarcinia and is also an anaerobic methanogenic archaeon. It uses CO_2_ and methyl acetate as substrates for methane production and is more competitive than the genus *Methanosarcina*, which also consumes methyl acetate, regardless of the concentration of acetic acid [22,55]. It has been shown that 60% of the methane produced by organisms and released into the atmosphere comes from methyl acetate [56], so it is thought that *Methanosaeta* is the main methane producer on the Earth [55]. *Methanosarcina* spp. are acetic acid trophic methanogenic bacteria that compete with *Methanosaeta* spp. for acetic acid while also producing methane and growing on formate, methanol, and hydrogen as substrates [57]. *Methanosaeta* and *Methanosarcinia* frequently combine [58], which may be one reason why both genera simultaneously have a significantly higher relative abundance of NHX2 than other sites. The relative abundance of Halobacterales was significantly higher in XHS2. It is a salt-loving archaea, generally considered to have evolved from salt-loving methanogenic archaea [59], and is commonly found in sea salt [60].

The comparison of LEFSE from the two lakes showed that NH (lake with birds) had five archaeal genera whose relative abundances were significantly different from those in XH (small lake). They are mainly concentrated in classifications under the phylum Halobacterota, such as the genera *Methanosarcina*, *Methanosphaerula*, and *Methanospirillum*. The latter two are both strictly anaerobic archaea that use hydrogen, carbon dioxide, or formate to produce methane [61,62]. Combined with Figure 2a, it can be seen that the relative abundance of the phylum Euryarchaeota of NH is significantly higher than that of XH, and the relative abundance of the genus *Methanobacterium* in this phylum varies significantly. The genus *Methanobacterium* uses H_2_/CO_2_ and formate as substrates to synthesize methane and also to syntrophically degrade formate [63]. SiO_3_^2−^, NO_3_^−^, NO_2_^−^, TOC, and MC were significantly different in the two lakes, and the RDA results show that TOC was correlated, and several other physicochemical factors were significantly correlated. However, the archaea with different abundances in the two lakes were concentrated in the methanogenic archaea type, so whether and how these physicochemical factors caused the differences in archaea remain to be investigated in a follow-up study. The two lakes have very different landscapes, with NH being the lake with a lot of bird habitats. Previous studies have shown that the archaeal diversity index increases with an increasing bird density [64], which is consistent with the results of this study. The results of the alpha diversity analysis in this study (Appendix A) also show that the archaea diversity in NH is higher than that in XH. The physiological activities of birds affect the soil environment [65] and thus the structure of archaeal communities, but there is a lack of studies in this area.

WGCNA is an R package, and its basic process is to first cluster highly correlated genes and form a module, summarize these clusters with hub genes, correlate the modules with each other and with out-of-sample features and calculate the correlation, and finally find the hub genes [66]. It is widely used in the biomedical field to analyze hub genes for diseases, such as aortic dissection [67], colorectal cancer [68], and endometrial carcinoma [69]. In this study, we used WGCNA to explore the correlations between highly correlated archaeal modules and physicochemical factors to identify significantly influential factors and hub archaea. The results showed that the turquoise and gray modules correlated strongly with a number of physicochemical factors, and the correlations of NH_4_^+^, SiO_3_^2−^, NO_3_^−^, and NO_2_^−^ could be corroborated with the RDA results. There were differences in the hub archaea genera compositions in different environmental sites, but *Woesearchaeales* was the hub genus for each site. For example, the hub archaea were *AR5* in NHS2, *Methanosaeta* in NHX2, and *Methanospirillum* in NHX3. The hub archaea of XHS were Candidatus_*Nitrososphaera* and *Methanoregula* of XHX, while the hub archaea of XHX3.3 had the same Candidatus_*Nitrososphaera* as XHS and *Methanobacterium*. Candidatus_*Nitrososphaera* is an ammonia-oxidizing archaea that uses NH_3_ and O_2_ for energy and has been shown to have the potential to use urea as a source of NH_3_ [70]. *Methanobacterium* is a hydrogenotrophic methanogenic archaea that uses CO_2_ and can also fix nitrogen [71,72]. A previous showed a significantly positive correlation between *Methanobacterium* and NH_4_^+^ [34]. Therefore, the differences in hub archaea in XHX3.3 may be caused by the significantly high NH_4_^+^ concentration at this site. *Woesearchaeales* are anaerobic archaea from the phylum Nanoarchaeota and are widely distributed in the environment, such as in Antarctic soils [41], trench sediments [73], sewage [74], and cows [75]. It has been shown that *Woesearchaeales* has positive correlations with *Methanoregula* and *Methanosarcina*, which have symbiotic or parasitic relationships with methanogens [24,74]. This may be an important reason why *Woesearchaeales* was identified as a hub archaeon.

In bacterial culture, a ‘sandwich agar plate method’ has been used, in which ‘helper’ bacteria are used as a medium sandwich to co-culture other marine bacteria [76]. Subsequent experiments have also shown the potential of the sandwich agar plate method for isolating uncultured bacteria [77]. Generally speaking, archaea are difficult to cultivate. Using bacteria as a reference, we would be able to co-cultivate more archaea with ‘helper’ archaea. However, the method of finding ‘helper’ bacteria through experimental testing is not applicable to archaea due to their noncultivability. Rapidly developing genomic technologies provide tremendous support for archaeal research [78]. However, many hypotheses about the evolution, physiology and diversity of archaea proposed by culture-independent methods have yet to be confirmed by culture experiments [79]. Improvements in traditional culture methods are therefore urgently needed. Therefore, if the hub archaea can be found through WGCNA, it may be possible to improve the traditional medium by virtue of its association with other archaea, so as to achieve the purpose of enriching culturable archaea. WGCNA was well used in our research to identify hub archaea.

## 5. Conclusions

It was found that significant differences in the archaea community structure exist not only between the intertidal zone and the subtidal zone in typical Arctic meltwater lakes, but also in lakes with different characteristics. According to the RDA analysis, NH_4_^+^, SiO_3_^2−^, MC, NO_3_^−^, and NO_2_^−^ were significantly and strongly correlated with the composition of the archaea community, as TOC and pH also had certain effects. Furthermore, WGCNA was used to explore the correlations among archaea modules and physicochemical factors. Then, the hub archaea were obtained. The sites with different environmental characteristics had different hub archaeal structures, but *Woesearchaeales* was the hub archaeon at each site. This study provides new ideas for future microbial diversity research and archaea culturability.

## Figures and Tables

**Figure 1 biology-12-01023-f001:**
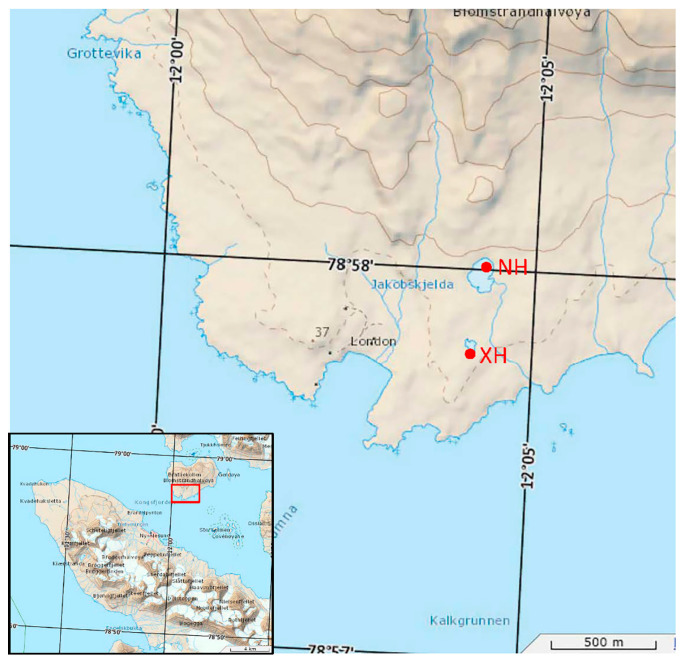
Map of sampling sites.

**Figure 2 biology-12-01023-f002:**
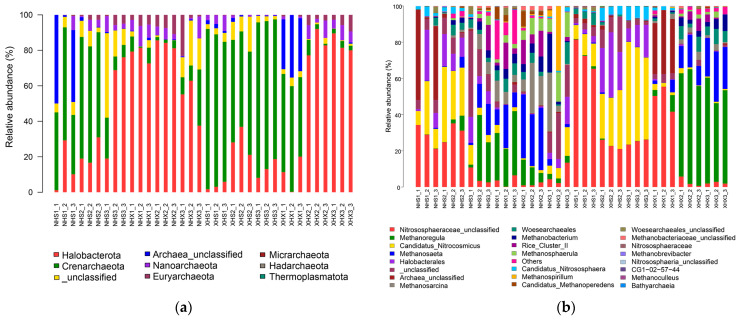
Bar chart of the relative species abundance of archaea at the phylum level (**a**) and genus level (**b**).

**Figure 3 biology-12-01023-f003:**
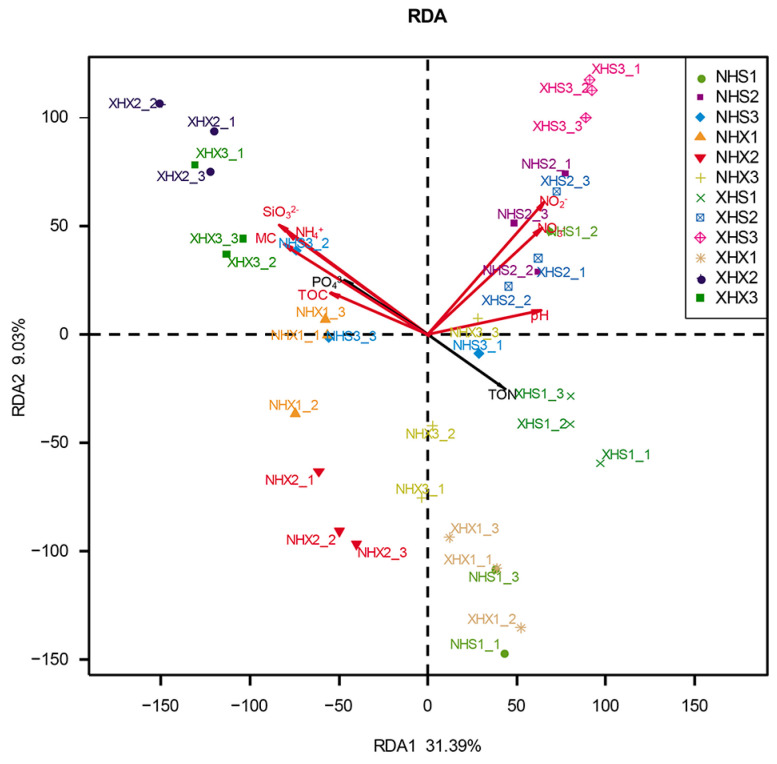
Redundancy analysis showing correlations between the composition of the archaeal communities and nine environmental factors for 36 samples from 12 sampling sites. The arrows represent the measured physicochemical factors.

**Figure 4 biology-12-01023-f004:**
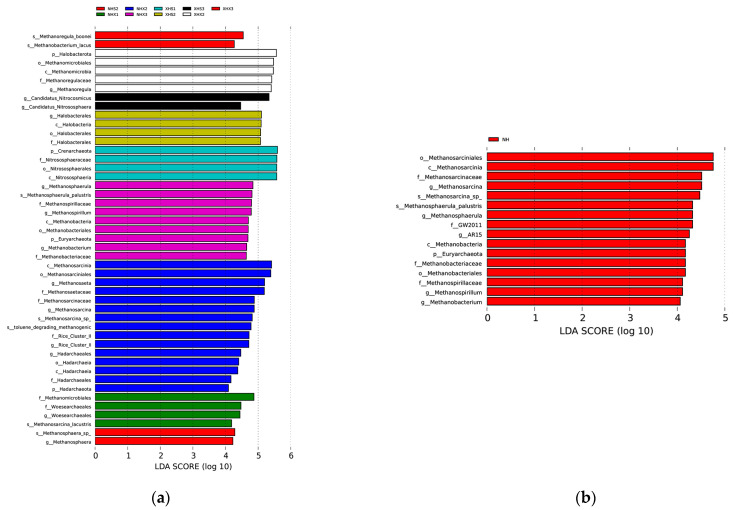
LDA scores showing archaeal taxa with significant different relative abundances. (**a**) shows the results among sampling sites and (**b**) shows the results between the two lakes.

**Figure 5 biology-12-01023-f005:**
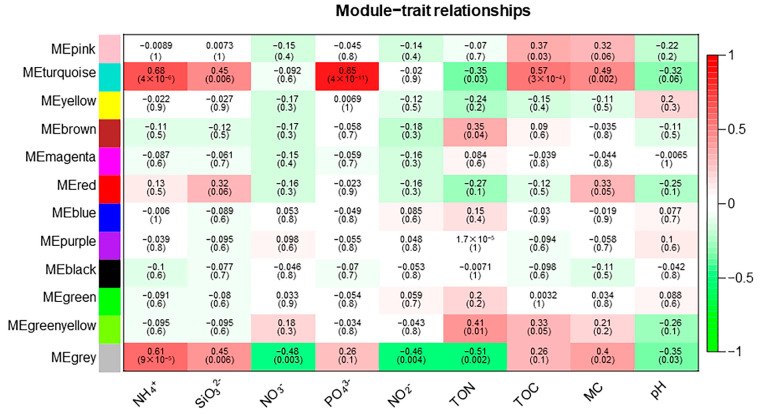
Relationships between archaeal modules and soil physicochemical factors. The horizontal coordinates are physicochemical factors and the vertical coordinates indicate modules. Red squares indicate positive correlations and green squares indicate negative correlations. The darker the color, the stronger the correlation.

**Figure 6 biology-12-01023-f006:**
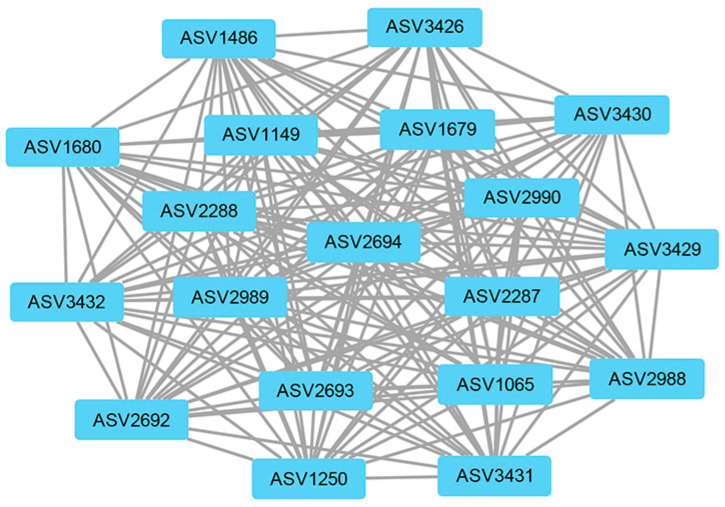
The network diagram of the turquoise module. The nodes represent ASVs, and the lines show the connections between them. The larger the size of a node, the more relevant it is to other nodes.

**Figure 7 biology-12-01023-f007:**
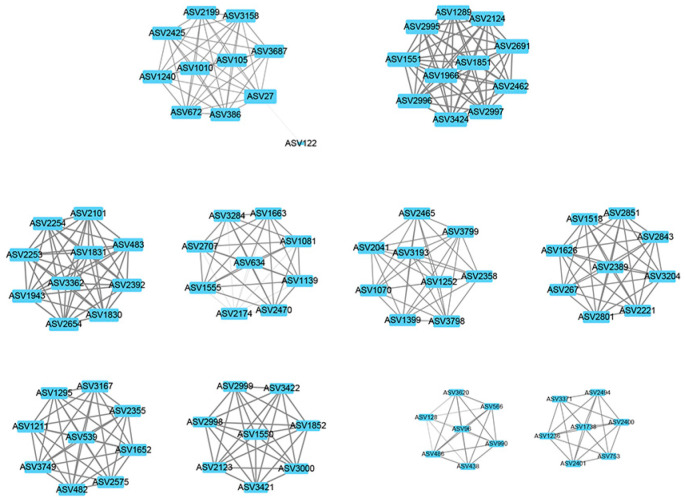
The network diagram of the gray module. The nodes represent ASVs, and the lines show the connections between them. The larger the size of a node, the more relevant it is to other nodes. The individual submodules are connected to each other by a number of ASVs with relatively low weight values.

**Table 1 biology-12-01023-t001:** Sample collection sites.

Sites	Location Profile	Coordinate
NHS1	Intertidal soil of the lake with birds	78.9656° N 12.0729° E
NHS2		78.9657° N 12.0701° E
NHS3		78.9662° N 12.0680° E
NHX1	Subtidal soil of the lake with birds	78.9656° N 12.0729° E
NHX2		78.9657° N 12.0701° E
NHX3		78.9662° N 12.0680° E
XHS1	Intertidal soil of the small lake	78.9631° N 12.0690° E
XHS2		78.9638° N 12.0678° E
XHS3		78.9630° N 12.0698° E
XHX1	Subtidal soil of the small lake	78.9631° N 12.0690°E
XHX2		78.9638° N 12.0678° E
XHX3		78.9630° N 12.0698° E

**Table 2 biology-12-01023-t002:** Physicochemical properties of soil samples.

Sites	pH	MC(%)	TOC (%)	TON (%)	NO_2_^−^-N (μg∙g^−1^)	NO_3_^−^-N (μg∙g^−1^)	NH_4_^+^-N (μg∙g^−1^)	SiO_3_^2−^-Si (μg∙g^−1^)	PO_4_^3−^-P (μg∙g^−1^)
NHS1	7.08 ± 0.22 fg	26.44 ± 1.36 c	6.74 ± 0.81 bcde	1.479 ± 0.116 bc	0.424 ± 0.138 bc	2.016 ± 0.740 c	3.249 ± 0.262 c	17.655 ± 0.635 c	0.049 ± 0.016 b
NHS2	7.68 ± 0.15 de	27.14 ± 1.77 c	4.23 ± 1.11 cde	1.052 ± 0.487 cd	0.587 ± 0.149 b	2.159 ± 0.021 bc	9.597 ± 1.901 c	16.852 ± 1.179 c	0.219 ± 0.034 b
NHS3	8.11 ± 0.15 ab	17.36 ± 2.37 d	1.08 ± 0.32 de	0.231 ± 0.064 e	0.173 ± 0.079 cde	0.998 ± 0.235 de	5.467 ± 4.379 c	20.347 ± 3.880 c	0.040 ± 0.023 b
NHX1	7.42 ± 0.15 ef	25.62 ± 2.88 c	8.10 ± 1.43 bcd	1.881 ± 0.328 ab	0.080 ± 0.043 de	0.764 ± 0.242 e	2.458 ± 1.417 c	14.402 ± 1.366 c	0.095 ± 0.060 b
NHX2	7.52 ± 0.14 de	15.58 ± 0.91 d	3.38 ± 1.29 cde	0.755 ± 0.321 de	0.039 ± 0.005 e	0.332 ± 0.070 e	1.423 ± 0.392 c	14.046 ± 0.619 c	0.074 ± 0.019 b
NHX3	8.10 ± 0.11 abc	14.06 ± 1.35 d	0.23 ± 0.10 e	0.132 ± 0.183 e	0.080 ± 0.038 de	0.364 ± 0.058 e	4.613 ± 2.351 c	27.733 ± 7.466 c	0.045 ± 0.048 b
XHS1	8.30 ± 0.15 a	12.36 ± 0.47 d	0.87 ± 0.48 de	0.207 ± 0.206 e	0.196 ± 0.078 cde	0.693 ± 0.187 e	2.318 ± 1.518 c	34.942 ± 4.522 c	0.028 ± 0.022 b
XHS2	7.76 ± 0.02 cde	31.29 ± 2.00 c	6.39 ± 0.52 bcde	1.736 ± 0.052 ab	0.590 ± 0.042 b	1.596 ± 0.158 cd	1.789 ± 0.379 c	19.337 ± 0.853 c	0.143 ± 0.228 b
XHS3	7.82 ± 0.08 bcd	17.60 ± 1.68 d	5.19 ± 3.13 bcde	1.289 ± 0.628 bcd	2.230 ± 0.430 a	5.975 ± 0.472 a	3.342 ± 0.239 c	12.103 ± 0.746 c	0.471 ± 0.189 b
XHX1	6.81 ± 0.07 gh	40.25 ± 6.82 b	12.27 ± 4.39 ab	2.232 ± 0.407 a	0.351 ± 0.101 bcde	2.767 ± 0.488 b	0.524 ± 0.257 c	10.295 ± 0.510 c	0.017 ± 0.008 b
XHX2	7.02 ± 0.36 gh	58.56 ± 2.92 a	9.64 ± 10.03 bc	0.320 ± 0.397 e	0.039 ± 0.022 e	0.609 ± 0.128 e	23.377 ± 18.074 b	191.481 ± 152.678 a	0.013 ± 0.010 b
XHX3	6.67 ± 0.23 h	62.43 ± 7.10 a	18.95 ± 4.56 a	0.135 ± 0.161 e	0.402 ± 0.259 bcd	1.612 ± 0.560 cd	65.510 ± 10.494 a	168.284 ± 53.541 b	18.857 ± 16.914 a

In a one-way ANOVA, significant differences between study sites were assumed at *p* < 0.05, and were followed by Duncan’s test. The letters a, b, c, d, e, f, g and h are used to show statistically significant differences.

**Table 3 biology-12-01023-t003:** Monte Carlo permutation test for the physicochemical factors and archaeal community.

	RDA1	RDA2	*r^2^*	*Pr* (>r)	
NH_4_^+^	−0.88670	0.46234	0.4339	0.001	***
SiO_3_^2−^	−0.88229	0.47071	0.4911	0.001	***
NO_3_^−^	0.76690	0.64176	0.4254	0.001	***
PO_4_^3−^	−0.91135	0.41164	0.1456	0.054	.
NO_2_^−^	0.71669	0.69739	0.5296	0.001	***
TON	0.89069	−0.45462	0.1322	0.100	.
TOC	−0.97268	0.23215	0.1773	0.039	*
MC	−0.91961	0.39284	0.4264	0.001	***
pH	0.95720	0.28943	0.2527	0.015	*

* Correlation is significant at the 0.05 level; *** correlation is significant at the 0.001 level.

## Data Availability

Not applicable.

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
