# Peer review of "Study of Archaeal Diversity in the Arctic Meltwater Lake Region"

_biology, 2023, doi:10.3390/biology12071023_

Round 1

Reviewer 1 Report

General Comments:

The study analyzed archaeal community structure and the influence of soil physiochemical factors in two typical meltwater lakes in Arctic Ny-Ålesund area. Differences in archaeal community structure were observed between the intertidal and subtidal zones of the two lakes. Several physicochemical factors played an important role in the diversity and structure of archaeal community by altering the abundance of certain hub archaea. The authors present an interesting study on the archaeal community structure in Arctic meltwater lakes in the Ny-Ålesund area.

However, as the authors mentioned that “the Arctic is experiencing significant temperature increasing under the global warming trend”, and “this study contributes to the research of archaeal communities in Arctic meltwater lakes in response to climate change”, I suggest the authors could supplement some discussion about the potential effect of climate change on archaeal community structure in Arctic lakes.

Specific comments:

Lines 26 and 35: As in general for the entire manuscript, all abbreviations have to be explained at first use.

Lines 70-71: I think the authors neglect the research of archaea in seawater (e.g., Jain and Krishnan, Antonie Van Leeuwenhoek, 2021; Zeng et al., Polar Biol, 2021) and marine sediment (e.g., Zeng et al., Front Microbiol, 2017) in Ny-Ålesund area.

Lines 82-89: I suggest that the whole paragraph could be placed into Discussion section.

Line 94: It should be 16S rRNA instead of 16SrRNA.

Line 99: What’s the purpose to narrow the range of “helper” archaea? To co-culture other archaea? Based on the present manuscript, I don’t think that the authors aim to cultivate archaea by finding “helper” archaea. I suggest to delete that sentence from the manuscript.

Materials and methods: There is no definition of intertidal soil and subtidal soil in “2.1 Study site description and sample collection”. It is actually important to help general readers understand the significance of the manuscript. In addition, I suggest the authors to provide a sampling map in the manuscript.

Lines 102-103: The repeated sentence should be deleted. “uniquely” can be deleted, too.

Table 2: Please supplement the explanation of different letters (i.e., from a to h) in the Table. Not all general readers are familiar with those letters. In addition, I don’t find the statistical analysis (e.g., one-way ANOVA and Duncan’s test) of soil physicochemical properties in Materials and methods section.

Line 172: Italicize all statistical symbols used in the manuscript. For example, p value (significance level).

Reviewer 2 Report

Overview

The manuscript “Study of archaeal diversity in the Arctic meltwater lake region” presents an investigation of the archaeal community of two lakes in the Ny-Alesund area in the Arctic. The study sites are interesting, and the research question (“What Archaea are living here and what factors influence their distribution?”) is good. 16S rRNA data and physicochemical data were collected, and statistical tests were used to identify significant differences in population structure and the role of key soil characteristics in influencing these communities.  Further information about the sample sites and the methodology needs to be provided, and some methods used were not appropriate (WGCNA) or were interpreted wrongly (LEFSE). With careful re-analysis and re-writing the paper should become publishable.

Concerns:

We need more details about the lakes and the samples that were collected

Which exact lakes are these? Can we have a map or diagram to show the lakes locations please. A photo of each lake would be really helpful too.

What are the features of these lakes?  Freshwater or saline? Are they permanent lakes or seasonal meltwater ponds? Are the ponds forming on a glacier surface? On or close to permafrost? Are they open to the ocean current, since they have an intertidal zone? 

What is the lake depth and profile – are the lakes stratified into oxic and anoxic zones? 

How did you select these two lakes? How far apart are they located? 

Have these specific lakes ever been researched before?  (such as for their bacterial or algal composition?) What are the species of note that live at the lake (birds, fish species, seagrasses etc). How close are the sites to human activities?

One site is described as “with birds” – what does this mean? Surely some birds may still visit the other lake – is there a quantified difference in the type or abundance of birds that visit each lake?

July 2018 is listed as when the samples were taken. Which days in July 2018? Were all samples taken on the same date? What time of day? What was the tide level when the intertidal soil samples were collected? Were all subtidal and intertidal samples taken at the same point in the tidal cycle (eg. Low tide)?   Were the samples for physicochemical analysis taken at the same times as those for the 16S analysis?

Please also list how many 16S sequences were obtained from each sample. This is very important for better understanding if your statistics were correctly applied, and will inform your conclusions. 

If you can provide more information about the sampling sites this will strengthen the paper and it could give more weight to your conclusions about why the communities are different. Anaerobic zones present in the lakes would make a lot of sense regarding all the methanogens you have detected?

More details needed in the methods section

Can you provide more details about what steps were performed in DADA2 – did you filter and trim reads? Did you check for and remove chimeras? What version of DADA2 did you use? Which database was used for assigning species identifications in DADA2? (Silva? RDP? Something else?). 

For the physicochemical data, what normalisations or transformations were applied to the datasets? Please give more details of what steps were performed in both QIIME and RDA. 

What version of LEFSE was used? Please give details of what steps (including normalisations, transformations etc) were performed in the methods. 

Much more care needs to be taken with results/interpretation

Be careful with your conclusions about the LEFSE data. Fig 3b shows 16 red bars, but this does not mean 16 different archaeal species. LEFSE is running comparisons at every taxonomic level (c = class, o = order, f = family, g = genus, s = species). So you only have 5 different species that were significantly different in their abundance in the two lakes if you consider their full taxonomy, namely:

d__Archaea; p__Halobacteriota; c__Methanosarcinia; o__Methanosarcinales; f__Methanosarcinaceae; g__Methanosarcina; s__Methanosarcina_sp_

d__Archaea; p__Halobacteriota; c__Methanomicrobia; o__Methanomicrobiales; f__Methanosphaerulaceae; g__Methanosphaerula; s__Methanosphaerula palustris

d__Archaea; p__Nanoarchaeota; c__Nanoarchaeia; o__Woesearchaeales; f__GW2011-AR15; g__GW2011-AR15; s__GW2011-AR15 sp000830295

d__Archaea; p__Halobacteriota; c__Methanomicrobia; o__Methanomicrobiales; f__Methanospirillaceae; g__Methanospirillum; s__Methanospirillum_sp_

d__Archaea; p__Methanobacteriota; c__Methanobacteria; o__Methanobacteriales; f__Methanobacteriaceae; g__Methanobacterium; s__Methanobacterium_sp_

Similarly with Fig 3a, the white bars represent a single organism (Methanoregula), the aqua bars represent a single organism (Nitrosospaeria) etc. Please correct the text describing the LEFSE results in lines 219 – 225 and in lines 335 – 352.

WGCNA is not suitable to make conclusions about the archaeal communities or to identify helper archaea.

Weighted Gene Correlation Network Analysis (WGCNA) is a clustering package which was created for transcriptome (RNA-seq) and microarray analysis. To create co-expression networks from RNA-seq or microarray data, a typical cut-off is 10 million reads per sample and 20 samples minimum. (https://doi.org/10.1093/bib/bbw139)

Although you have 36 samples in this study, you only have 1,960,571 archaeal 16S sequences in total. This is not sufficient to use this tool and get meaningful results. You can see this in your Fig 5 and Fig 6 - In these figures all the nodes appear equally connected. There is no true hub node (or hub species) being identified here (See Fig 2 of the article referred to above, which shows how hubs should appear to be like a highway between different regions of the graph). WGCNA is more appropriate to use on shotgun metagenome sequencing datasets (if sufficiently large!), rather than 16S amplicon sequencing as you have in this manuscript.

All sections relating to WGCNA should be removed from the paper. (Lines 98, 154 – 157, 354 – 379, 395 – 399, and Figures 4,5 and 6.)

I agree that it is desirable to identify key species which influence and support the growth of the archaeal community (which is your stated aim for using WGCNA). It would be wonderful to find helper species that could be co-cultured in order to get more environmental archaea growing in laboratory culture.  You should use phyloseq on your dataset after the DADA2 steps – phyloseq is an R program with specific tutorials for analysing microbial communities and their associated environmental data. If phyloseq finds a strong association between Woesarchaeales and the other taxa in your sample sites then this may be a step forwards to identifying them as helper archaea. 

Minor points:

Line 136- 137 says  “V3-V4 region of the 16S gene” however primers Arch519F and Arch915R amplify the V4-V5 region. If you used Arch519F/Arch915R please change the wording to “V4-V5 region”, or please update with the primers you used to amplify V3-V4. 

Line 145: Serial Read Segment Archive. I think you mean Sequence Read Archive (SRA)?

Line 145-147 The DADA2 method in the QIIME 2.0 system was used for noise reduction to perform dereplication or equivalent to clustering at 100% similarity instead of clustering at similarity

This is unclear. What similarity did you cluster to?  

Conclusion: I think you have an interesting sample site and dataset. The paper is not acceptable at present but if the study sites and methods are described more fully, and appropriate analysis techniques (not WGCNA) are used then the paper should become acceptable for publication.

Please have an English speaker proofread. 

Take care with the archaeal taxonomy and when discussing results at the phylum/class/genus/species levels. 

Round 2

Reviewer 1 Report

I think the revised manuscript is ready for publication.